# The Thioredoxin System of Mammalian Cells and Its Modulators

**DOI:** 10.3390/biomedicines10071757

**Published:** 2022-07-21

**Authors:** Aseel Ali Hasan, Elena Kalinina, Victor Tatarskiy, Alexander Shtil

**Affiliations:** 1T.T. Berezov Department of Biochemistry, Peoples’ Friendship University of Russia (RUDN University), 6 Miklukho-Maklaya Street, 117198 Moscow, Russia; ali.aseel.hasan@gmail.com; 2Laboratory of Molecular Oncobiology, Institute of Gene Biology, Russian Academy of Sciences, 34/5 Vavilov Street, 119334 Moscow, Russia; tatarskii@gmail.com; 3Laboratory of Tumor Cell Death, Blokhin National Medical Research Center of Oncology, 24 Kashirskoye Shosse, 115478 Moscow, Russia; shtilaa@yahoo.com

**Keywords:** thioredoxin system, structure and catalytic function, inhibitors and activators, redox regulation, apoptosis

## Abstract

Oxidative stress involves the increased production and accumulation of free radicals, peroxides, and other metabolites that are collectively termed reactive oxygen species (ROS), which are produced as by-products of aerobic respiration. ROS play a significant role in cell homeostasis through redox signaling and are capable of eliciting damage to macromolecules. Multiple antioxidant defense systems have evolved to prevent dangerous ROS accumulation in the body, with the glutathione and thioredoxin/thioredoxin reductase (Trx/TrxR) systems being the most important. The Trx/TrxR system has been used as a target to treat cancer through the thiol–disulfide exchange reaction mechanism that results in the reduction of a wide range of target proteins and the generation of oxidized Trx. The TrxR maintains reduced Trx levels using NADPH as a co-substrate; therefore, the system efficiently maintains cell homeostasis. Being a master regulator of oxidation–reduction processes, the Trx-dependent system is associated with cell proliferation and survival. Herein, we review the structure and catalytic properties of the Trx/TrxR system, its role in cellular signaling in connection with other redox systems, and the factors that modulate the Trx system.

## 1. Introduction

Redox-dependent proteins play a significant part in maintaining redox homeostasis and the redox-dependent regulation of cellular processes, including proliferation/differentiation and apoptosis [1,2,3]. In the antioxidant system of cell protection, along with key antioxidant enzymes, an important role is played by the thioredoxin (Trx)-dependent system that is involved in the processes of cellular redox-dependent regulation through the control of thiol–disulfide exchange [4,5]. The combination of antioxidant properties and the ability to activate the transcription of genes, including some antioxidant enzymes, as well as to inhibit redox-dependent pathways of apoptosis activation, indicates an important contribution of this system to the antioxidant defense system, which increases the resistance of cells to oxidative stress [6].

The Trx-dependent system includes disulfide reductase Trx and thioredoxin reductase (TrxR), which uses NADPH(H+) as a co-substrate. Trx is necessary for the reduction of disulfides, particularly the oxidized form of peroxiredoxins, which catalyze the reduction of H_2_O_2_ to water and enhance the mechanism of controlling the cellular reactive oxygen species (ROS) level. As a result of the reduction of the oxidized substrate with the participation of Cys^32^ and Cys^35^ in the active center of the dithiol form of Trx, its oxidized form is formed, the reduction of which is carried out by TrxR, which also makes an independent contribution to the antioxidant potential of the cell by reducing lipid hydroperoxides [7].

Over the past ten years, numerous studies have been devoted to the role of the Trx-dependent system in oncogenesis. The overexpression of both cytoplasmic and mitochondrial TrxR isoforms (TrxR1, TrxR2) has been found in various types of malignant neoplasms, including breast, lung, oral cavity, and squamous cell carcinoma [8,9]. The hyperexpression of Trx isoforms in tumor cells is aimed at utilizing the excess amount of ROS produced by them and is closely related to the degree of tumor growth [10,11]. The Trx-dependent system plays a significant role in the activity of redox-dependent signaling, the change in the state of which in tumor cells is still poorly understood.

Herein, we review the structure and catalytic properties of the Trx/TrxR system, its role in cellular signaling in connection with other redox systems, and the factors that modulate the Trx system.

## 2. Trx/TrxR System: Structure and Functions

### 2.1. Trx1 and Trx2 Isoforms: Structure and Functions

In mammals, the Trx system is important for protection from the effects of ROS [12]. Trx folds with a redox-active C-X-X-C motif form a superfamily of proteins associated with thiol–disulfide exchange that involves disulfide bridge formation. This mechanism has been found in all organisms [7,13]. Proper oxidative protein folding within cells is essential for protein stability and function, and misfolding leads to severe diseases. Trx, the most representative member of this family, is a 12 kDa protein found in bacteria, plants, and animals [14].

The three-dimensional structures of Trxs from different species (*E. coli* Trx family and *Chlamydomonas reinhardtii*) have been revealed by *X*-ray crystallography [15,16], and the structures of both oxidized and reduced Trx states from *Ehrlichia chaffeensis* have been elucidated by NMR in solution [17].

The basic Trx fold motif is made up of three α-helices surrounding a central core made of four β-sheets [18,19]. In addition to the basic fold, Trx itself has an extra β-sheet and α-helix at the N-terminus. In the core region, five β-strands are flanked by four α-helices. The β-sheets and α-helices of the Trx fold can be subdivided into N-terminal β_1_α_1_β_2_α_2_β_3_ and C-terminal β_4_β_5_α_4_ motifs connected by a loop of residues involving the α_3_-helix. The β-strands of the N-terminal motif run in the same direction, whereas the two β-strands of the C-terminal motif are anti-parallel to each other. The α_2_ and α_4_ helices line up in a parallel fashion in one direction on the sheet while the α_3_ runs along the opposite face of the β-strands and is perpendicular to the other helices [20,21].

In mammalian cells, two major Trx types have been characterized that differ in their intracellular localization, tissue-specific expression patterns, and subcellular structure: cytosolic Trx-1 and mitochondrial Trx-2 [18]. Although Trx-1 is mainly located in the cytosol, it can migrate to the nucleus upon nitrosative/oxidative stress [22] or be secreted out of the cell [23]. Trx-2 is the vital mitochondrial redox isoform [24]. Both Trx-1 and Trx-2 are characterized by a conserved disulfide active site sequence Trp-Cys-Gly-Pro-Cys (WCGPC) [25].

The two residues Cys^32^ and Cys^35^ at the active site (essential for Trx activity and conformation) are readily oxidized and undergo a reversible redox reaction between an oxidized disulfide and a reduced dithiol [18]. Other conserved residues are not strictly required for the activity but dictate the thermodynamic and redox properties of the protein [20].

The proposed mechanism for the reaction of protein disulfide reduction is as follows: first, the reduced Trx binds non-covalently to an oxidized disulfide-containing protein substrate via a conserved hydrophobic surface area surrounding the Trx active site [26]. The pKa ~7 of the N-terminal active site cysteine is substantially lower than the pKa of free cysteine residues in solution [27]. Under physiological conditions, a large fraction of the sulfur in the N-terminal cysteine is present as a thiolate, a reactive deprotonated form of thiol. This thiolate can act as a nucleophile to interact with a variety of substrates, leading to the formation of an intermolecular mixed disulfide (Trx–*S*–*S*–protein) and releasing a free thiol [28]. In contrast, the high pKa ~9 of the C-terminal cysteine promotes its existence as a thiol. The C-terminal thiol must be activated as a thiolate to facilitate the next step of the reaction where the fully reduced target protein and the disulfide-containing Trx are generated [20,26]. Oxidized Trx is reduced to its active state by electrons from NADPH(H+). The reaction is catalyzed by TrxR and enables the onset of the next reaction cycle [20].

Besides two cysteine residues in the Trx-1 active site, the mammalian Trx-1 contains additional conserved cysteine residues outside their active site (at positions 62, 69, and 73 of human Trx-1) that are not found in mammalian mitochondrial Trx-2 or in Trx from other species [25,29]. Cys^62^ and Cys^69^ are buried in the protein interior and lie at either end of a short α_3_-helix and Cys^73^ is on a hydrophobic patch on the protein surface [30]. These additional cysteine residues are involved in the activity of mammalian Trx depending on their redox state; for example, the non-active site disulfide formed between Cys^62^ and Cys^69^ inhibits Trx-1 activity for redox signaling under oxidative stress conditions. The *S*-nitrosylation of Trx at Cys^69^ under basal conditions is required for ROS scavenging and to preserve the redox regulatory activity, thereby contributing to the protein’s anti-apoptotic functions [31].

Along with participation in oxidation-reduction pathways, Trx possesses chaperone-like properties that are important for protein folding and renaturation after stress. Trx promotes the functional folding of citrate synthase and α-glucosidase after urea denaturation and galactose receptor folding in *E. coli* [32]. In vitro, both tobacco plastid isoforms Trxs *f* and *m* facilitate the reactivation of the cysteine-free form of chemically denatured glucose-6 phosphate dehydrogenase and prevent the thermal aggregation of malate dehydrogenase [33]. The cytosolic soybean Trx revealed the same activity as a molecular chaperone for peroxisome matrix proteins [34].

Recently, it has been reported that Trx-2 from *Trypanosoma brucei* acts as a molecular chaperone to prevent protein aggregation induced by temperature-mediated structural changes [35]. In comparison to Trx-1, detailed information on the structure and function of Trx-2 is relatively scarce. The crystal structure of full-length Trx-2 from the multi-stress resistant bacterium *Deinococcus radiodurans* (DrTrx-2) has been elucidated. Trx-2 showed an N-terminal extension that forms a zinc finger domain with two CXXC motifs [36].

### 2.2. TrxR1 and TrxR2 Isoforms

TrxR (EC 1.6.4.5) is a homodimeric selenocysteine-containing flavoprotein that catalyzes the NADPH-dependent *reduction* of thioredoxin [37]. Trx and TrxR form the Trx antioxidant system for maintaining cellular redox homeostasis [4,6]. Around 55 kDa for each TrxR subunit has been identified primarily in mammals, and a TrxR of 35 kDa for each subunit is present in bacteria, plants, archaea, and most unicellular eukaryotes [38]. In the mammalian TrxR, each monomer includes FAD as a prosthetic group, an NADPH binding site, and a redox active site containing a dithiol/disulfide motif. The human placental TrxR has been purified and cloned and shows only 31% sequence identity with prokaryotic TrxRs [39].

The catalytic site -Cys-Val-Asn-Val-Gly-Cys- of the human TrxR is located in the FAD domain, whereas the respective site of *E. coli* TrxR, -Cys-Ala-Thr-Cys-, is part of the NADPH domain [40,41]. The C-terminus of the mammalian TrxR has a conserved Gly-Cys-SeCys-Gly containing a unique and important SeCys as another catalytic active site that is essential for the reduction of Trx and other substrates, including glutaredoxin 2 (Grx-2), protein disulfide isomerase, selenite, vitamin C and cytochrome c, and drugs such as motexafin, gadolinium, and alloxan. The broad substrate specificity of the mammalian TrxR is due to its flexible C-terminal tail and the high reactivity of the SeCys pair that is not found in its bacterial counterparts [42].

TrxR has three mammalian isoforms: cytosolic and nuclear TrxR-1, mitochondrial TrxR-2, and TrxR-3 (also known as thioredoxin glutathione reductase, TGR), the latter of which is expressed only in the testes [43,44]. These isoforms are encoded by three separate genes, *TNXRD1*, *TNXRD2*, and *TNXRD3*, respectively. TrxR-1 and humanTrxR-2 are closely related, displaying 56% identity and 84% similarity to the primary amino acid sequence. However, TrxR-2 differs from TrxR-1 by the presence of a 33-amino acid extension at the N-terminus that has the characteristic properties of the mitochondrial translocation signal [45].

The first step of the reductive half reaction of TrxR includes the reduction of FAD by NADPH in one subunit, and then FAD subsequently transfers the reducing equivalents to the N-terminus (Cys-Val-Asn-Val-Gly-Cys), in which Cys^59^ and Cys^64^ form an active site motif, and reduces the disulfide to a dithiol pair. This N-terminal dithiol pair further gives electrons to the C-terminal selenenylsulfide (Cys^497^–SeCys^498^) of another subunit and reduces it to a selenolthiol pair. This reduced C-terminal selenolthiol pair acts as a second redox center; electrons are transferred from the redox-active disulfide via the redox center at the C terminus to TrxR substrates such as oxidized Trx, glutaredoxin 2, protein disulfide isomerase, and small molecules, e.g., selenites, hydrogen peroxide, dehydroascorbate, lipoic acid, ubiquinone, cytochrome c, alloxan, and motexafin gadolinium [43].

### 2.3. Extracellular Trx/TrxR

Trx-1 is mainly localized in the cytosol but can be secreted out of the cells in two forms, a full-length Trx-1 and a truncated form called Trx-80. The latter lacks redox activity but stimulates peripheral blood mononuclear cells (PBMC) [46,47]. Although Trx lacks a classical signal peptide, it is exported to the extracellular environment via the ER/Golgi-independent pathway (also called unconventional or non-classical secretion). The precise mechanism of Trx secretion is unclear.

To date, no specific cell surface receptors for Trx have been identified [14,48]. Trx-1 is an autocrine growth factor for human T-lymphotropic virus-1 and Epstein–Barr virus (EBV)-transformed B lymphocytes and also acts as a cytokine and chemokine for immune cells [49]. Increased levels of extracellular Trx have been reported in many pathological conditions associated with oxidative stress. For instance, extracellular thioredoxin (Etrx3/REQ_13520) is essential for the resistance of *Rhodococcus equi*, an actinobacterial pathogen, to oxidants [50]. There is increasing evidence that extracellular Trx also plays a role in the immune response due to its ability to selectively recognize the C46–C99 disulfide of IL-4, thereby inactivating the cytokine activity in TF-1 erythroleukemia cells [51].

Extracellular Trx stimulates tumor cell proliferation [52]. Mechanisms presume an increased cytokine production (IL-1, IL-2, and TNFa), as well as the stimulation of growth factors and proliferation-associated transcription factors [53].

Additionally, Trx has been identified as a lipid raft (LR)-associated protein [54]. LRs, the plasma membrane microdomains, contain sphingolipids and cholesterol [55] that are able to form membrane macrodomains, control the redox state of cell surface molecules, and influence the downstream signaling pathways [14]. In particular, a Trx-C35S mutant in which Cys^35^ of the active site was replaced with serine, was quickly bound to the cell surface and internalized in a LR-dependent manner. This suggests that the cysteine residue in the Trx active site plays a fundamental role in the internalization of extracellular Trx through LR [56].

### 2.4. Cytosolic and Mitochondrial Trx/TrxR. Role in Apoptosis Mechanism

Apoptosis signal-regulating kinase 1 (ASK1) belongs to the mitogen-activated protein kinase (MAPK) family that phosphorylates and activates both c-Jun N-terminal kinase (JNK) and p38/MAPK pathways [57]. Reduced Trx-1 binds to the N-terminal regulatory domain of ASK-1 through its redox-active site that is modulated by oxidative stress [58]. The inhibition of Ask1 oxidation via the overexpression of Trx-1 impairs JNK activation and apoptosis [59].

A single Trx-1 cysteine (Cys^32^ or Cys^35^) is required to induce ASK-1 ubiquitination and degradation. Recent results have revealed that the modification of Cys residues in human cytosolic Trx-1 by *p*-benzoquinone leads to the dissociation of the Trx1–ASK-1 complex, with subsequent activation of ASK1, the p38/MAPK pathway, and apoptosis [60]. Additionally, TAT-2GTP1, a cell-permeable derivative of the biotinylated 2GTP1 peptide, selectively disrupts the Trx1–Ask1 interaction that induces phosphorylation and the subsequent activation of ASK1, leading to JNK activation and reduced viability of cancer cells [61]. Trx-2 associates with mitochondrial ASK1 and activates the JNK-independent apoptosis pathway. Trx-1 and Trx-2 bind to Cys^250^ and Cys^30^ in the N-terminal regulatory domain of ASK-1, respectively [62]. The Trx-2–ASK-1 signaling pathway plays a regulatory role in mitochondria-induced apoptosis during the progression of *Pemphigus vulgaris* [63].

The final actor in this triad is Trx interacting protein (Txnip) (also called Trx binding protein-2; TBP-2), which binds specifically to reduced Trx and can function as a potent negative Trx regulator. The interaction between Trx and Txnip involves disulfide bond formation between the reduced Trx and Txnip Cys^247^ [64]. This interaction allows for the breakdown of the Trx-1–ASK-1 complex and the reactivation of *ASK1* activity, which in turn induces apoptosis through JNK and p38 cascades [65]. The Trx system is regulated by the ASK-1/JNK/p38/survivin apoptosis pathway during testicular ischemia reperfusion injury (tIRI) [66].

Caspases, the executors of apoptosis, comprise a family of cysteine proteases [67]; that is, their activity depends on cysteine in their active site. Trx can suppress apoptosis by catalyzing the S-nitrosation of procaspase-3 and caspase-3 in Jurkat cells [68]. It has been demonstrated that under physiological conditions, reduced Trx-1, but not Trx-2, interacts via its active site cysteines with apoptosis inducing factor (AIF) to suppress AIF-mediated DNA damage by modulating AIF–DNA interaction, whereas under oxidative stress conditions, the interaction between Trx-1 and AIF is disrupted [69].

### 2.5. Nuclear Trx Function

Under nitrosative/oxidative stress conditions, Trx-1 migrates into the nucleus. This phenomenon is involved in cell survival [22]. Similarly to ROS, reactive nitrogen species (RNS) can induce nitrosative damage. Nitric oxide (NO) is synthesized via the oxidation of L-arginine by three NO synthase isoforms: endothelial (eNOS), inducible (iNOS), and neuronal (nNOS) [70]. NO has emerged as an essential regulator of several cellular functions including blood coagulation, inflammation, and cell adhesion. Additionally, NO has been implicated in neurodegenerative diseases and cancer [71].

The mechanism of Trx-1 nuclear migration under nitrosative stress conditions evoked by a nitrosothiol, S-nitroso-N-acetylpenicillamine (SNAP), is strongly associated with the p21Ras-ERK1/2 survival signaling pathway. As a result of the nitrosylation of p21Ras and the activation of ERK1/2 MAP kinases, Trx-1 is accumulated in the nucleus through the down-regulation of Txnip. Trx1 nuclear translocation activates the transcription factors related to cell survival and proliferation [22,72].

## 3. Inhibitors and Activators of the Trx/TrxR System

### 3.1. TrxR Inhibitors

As is analyzed below, Trx/TrxR overexpression has been observed in many malignancies. Thus, there is increasing interest in the development of Trx/TrxR inhibitors that have potential anticancer activity [73,74]. Zhang and co-workers presented four classes of TrxR inhibitors: (1) metal-containing inhibitors (Au, Pt, Sn, Ru, Rho, La, Si, Fer); (2) natural products and their synthetic analogues (phenylpropanoids and polyphenols, quinone compounds, terpenoids, nitrosoureas, and chromenes); (3) Se-, S-, Te-, and As-containing compounds; (4) miscellaneous inhibitors [75,76]. The chemical structures of inhibitors and half maximal inhibitory concentrations (IC_50_) are summarized in Table 1 and Table 2.

#### 3.1.1. Inhibition of TrxR by Metal Complexes

According to the hard and soft acids and bases (HSAB) theory, the thiol and selenol groups (‘soft base’) in the side chains of Cys and Sec have a high affinity for metal complexes (‘soft acid’) and yield various metal complexes that are potent Trx/TrxR inhibitors [75]. Among the mammalian TrxR inhibitors, gold(I) complexes are the most potent inhibitors reported thus far [92]. As is shown in Table 1, these inhibitors have a strong effect against TrxR, with IC_50_ values in the nanomolar range, but are less effective in comparison with auranofin.

Among the metal inhibitors, auranofin (AF), a gold-containing compound, is classified by the WHO as an anti-rheumatic agent. As an anticancer drug [93], auranofin triggers apoptosis via the up-regulation of death receptors and caspase activation in Hep3B hepatocellular carcinoma cells [94]. Mitochondrial TrxR-2 represents an attractive target for auranofin, which causes mitochondrial dysfunction in cancer cells [76]. Platinum-containing drugs (PtDs) cisplatin, carboplatin, and oxaliplatin are other examples of effective metals for TrxR suppression. The inhibition of mammalian TrxR-1 by PtDs is accompanied by the transcriptional activation of Nrf-2-regulated genes including *TRXRD1* [95]. Compared with Pt, Au anticancer compounds are strikingly more effective inhibitors of recombinant TrxR-1 [96]. For instance, the IC_50_ values of auranofin are 0.12 and 3.17 µM for HeLa and MRC-5 cells (Table 2), respectively. These values are lower for cisplatin (11.5 and 7.9 µM, respectively) [97].

Mercury-containing organic compounds inhibit both Trx and TrxR activities in vitro. In vivo results showed that methylmercury (MeHg) inhibited Trx and TrxR in the brain and liver of experimental zebra sea breams. These results indicated that the Trx system could be a target for the toxicity of MeHg, with TrxR being particularly affected [79]. 

#### 3.1.2. Mechanisms of Trx/TrxR Inhibition by Natural and Synthetic Compounds

Traditional therapy using natural products has long been used for anticancer, antioxidant, and anti-inflammatory purposes, involving flavonoid products, of which curcumin, myricetin, and quercetin are among the best known examples [98]. Curcumin was found to be an antioxidant and anticancer agent by irreversible covalent modification of the redox-active residues Cys^496^ and Sec^497^ in TrxR [99]. Combinations of curcumin and quercetin modulate Wnt/β-catenin signaling and induce apoptosis in A375 melanoma cells [100].

Myricetin and quercetin irreversibly inhibit TrxR (IC50 values of 0.62 and 0.97 μmol/L, respectively) and arrest the growth of a lung cancer cell line. The inhibition of TrxR was related to time exposure to the inhibitors, the concentration of NADPH(H+), and the amount of dissolved oxygen [101].

The natural diterpenoid isoforretin A (IsoA) has been shown to effectively inhibit Trx-1 by conjugation to Cys^32^ and Cys^35^ residues in the catalytic sites of Trx-1. IsoA induces intracellular ROS burst and apoptosis in hepatocellular carcinoma [102].

Indolequinones have exhibited potent antitumor activity, with growth inhibitory IC_50_ values in the low nanomolar range (Table 2). The compound 5-methoxy-1-methyl3-[(2,4,6-trifluorophenoxy)methyl]indole-4,7-dione was found to induce time- and concentration-dependent apoptosis and to be a potent inhibitor of TrxR-1 in MIA PaCa-2 cells at concentrations equivalent to those that induce growth-inhibitory effects [83].

#### 3.1.3. Trx/TrxR Inhibition by Organochalcogen and Organoarsenic Compounds

Organochalcogen compounds contain S, Se, and Te or a dichalcogenide bond (-S-S-, -Se-Se-, or -Te-Te-) [103]. Diselenides and ditellurides have similar chemical properties to disulfides. The catalytic process of the thioredoxin system involves the essential thiol–disulfide and selenolthiol–selenenylsulfide exchange reactions, and these exchange reactions are readily intervened in by dichalcogenides, leading to the inhibition of TrxR or Trx [76]. Under physiological conditions in vivo, diselenides (RSeSeR) can be reduced to form selenol/selenolate intermediates (RSeH/RSe−) via NAPDH oxidation by a reaction catalyzed by TrxR. Diselenoamino acid derivatives can mimic GPx and be a substrate for mammalian TrxR [104].

### 3.2. Activators of the Trx/TrxR System

*TRX* gene expression can be induced by natural substances such as estrogens, prostaglandins, and cAMP. Geranylgeranylacetone (GGA), a Trx inducer agent, is a natural product that is derived from a plant source and used as an anti-ulcer drug. GGA protects cells through Trx induction at the mRNA and protein levels and the activation of NFκB and AP-1 transcription factors [105]. Inducible Trx-1 expression and GGA can protect from morphine effects in mice [106].

Selenite can be used to recover TrxR activity and cell viability inhibited by HgCl_2_ or MeHg. Treatment with selenite and NADPH led to almost full recovery of TrxR inactivated by HgCl_2_. The mechanism seems to be due to the reduction of selenite to selenide, which can remove the mercury from the selenoenzyme TrxR active site to generate mercury selenide [107].

## 4. Trx/TrxR Functions in Health and Disease

Oxidative stress is a condition that refers to the increased production and accumulation of ROS which are natural byproducts of aerobic respiration and energy extraction [108]. These include free radicals such as superoxide anions (O_2_^•−^), hydroxyl radicals (^•^OH), and non-radical molecules such as hydrogen peroxide H_2_O_2_ and singlet oxygen ^1^O_2_, which all constitute partly reduced forms of molecular oxygen (O_2_) [109]. Under physiological conditions, low basal ROS levels are produced by mammalian cells to mediate diverse physiological responses, including growth, migration, and differentiation. However, ROS excess can damage DNA, proteins, and lipids and lead to cell death, cancer, and/or senescence. In order to avoid or reverse ROS-induced damage to macromolecules, proper redox conditions must be maintained [110].

Multiple antioxidant defense systems have evolved to protect against lethal accumulation of ROS in the cell, including the Trx system, glutathione, and enzymatic ROS scavengers (e.g., glutathione peroxide and glutathione-*S*-transferases). Both enzymatic and non-enzymatic antioxidants are regulated by the common transcription factor nuclear factor Nrf2, which is translocated to the nucleus in response to oxidative stress (Figure 1) [111].

Active phosphorylated Nrf2 is transferred to the nucleus to affect the transcription of genes encoding antioxidant proteins and enzymes such as glutathione peroxidase 1 (*GPX*1), glutathione S-transferase mu 1 (*GSTM1*), glutamate–cysteine ligase catalytic subunit (*GCLC*), glutathione reductase (*GSR*), ferrochelatase (*FECH*), *TRX*, *TXNRD1*, and NAD(P)H quinone dehydrogenase 1 (*NQO1*) [112]. Many reports have shown that Nrf2 is able to regulate the redox-regulated enzymes such as heme oxygenase-1 (HO1) via the activation of extracellular regulated kinase (ERK) and phosphatidylinositol 3-kinase (PI3K/Akt) signaling [113]. Additionally, AMPK activation directly phosphorylates Nrf2 at Ser^550^ in vivo and at Ser^558^ residue in vitro, which, in conjunction with AMPK-mediated glycogen synthase kinase 3β (GSK3β) inhibition, facilitates the nuclear accumulation of Nrf2 for the antioxidant response element (ARE)-mediated gene transcription. Besides its role in the activation of Nrf2, the PI3K/Akt pathway causes the inhibitory phosphorylation of GSK3β [114]. Taken together, the Nrf2 target genes can be divided into different groups, as depicted in Figure 1.

The Trx/TrxR system plays an important role in the regulation of Nrf2 activity. Nuclear Trx1/Ref-1 is important for the reduction of critical Cys residues in Nrf2: one is important for DNA binding and the other is involved in nuclear export [115]. Furthermore, some reactive molecules that target TrxR1 may not only inhibit the enzyme but also transform the protein to pro-oxidant SecTRAPs (selenium compromised thioredoxin reductase-derived apoptotic proteins) with NADPH oxidase activity, thus further promoting the activation of Nrf2 in any cells that survive such an oxidative challenge [116]. The DNA binding activity of transcription factors NF-kB, AP-1, p53, and the glucocorticoid receptor is also regulated by the Trx1-reducing activities of essential cysteine residues [22].

As mentioned above, redox signaling is essential for controlling cell fate by the Trx system [117], so it is not surprising that this system has been implicated in cancer biology [118]. Elevated expression of Trx/TrxR has been detected in multiple human tumor types such as breast, thyroid, prostate, and colorectal carcinoma, and melanoma where it is associated with aggressive behavior [74,119]. 

The Trx/TrxR system also plays a crucial role during carcinogenesis, including the promotion of proliferation and tumor growth. Tumor cells transfected with Trx cDNA show increased growth and decreased apoptosis, while cells transfected with a redox-inactive Trx mutant display attenuated growth [120].

In cardiovascular disorders associated with oxidative stress, Trx-1 scavenges ROS and exerts a protective role to maintain cellular redox balance [121]. Under conditions of hypoxic/ischemic stress, Trx-1 effectively aids wound healing through improved angiogenesis, as well as increased capillary density and cell proliferation in a murine ischemic wound model. Experimental data demonstrate that Trx-1 therapy at the ischemic wound modulates the expression of pro-angiogenic genes by activating the PI3K/Akt survival pathway followed by GSK-3 β-inhibition and β-catenin translocation to the nucleus. Nuclear β-catenin binds to the T-cell factor/lymphoid enhancer factor (TCF/LEF) family of transcription factors and triggers the expression of angiogenic genes such as the vascular endothelial cell growth factor (VEGF) gene. VEGF subsequently binds to its receptor Flk-1 and activates the p38-MAPK cascade for migration and survival [122].

Similar results have been obtained in a number of cell cancer lines (MCF-7 human breast, HT29 human colon carcinomas, and WEHI7.2 mouse lymphoma) transfected with human Trx-1. The human Trx-1 increases HIF-1-α levels, VEGF production, and tumor angiogenesis. In contrast, transfection with a redox-inactive Trx-1 mutant (Cys^32,35^/Ser^32,35^) markedly decreased HIF-1α and VEGF in MCF-7 cells [123].

Extensive overlaps have been reported between the Trx and glutaredoxin systems. It has also been reported that the GSH/Grx system has a backup role in reducing Trx-1 in TrxR-1-deficient HeLa cells [124]. Both systems act as antioxidant regulators in response to oxidative/nitrosative stress. Increased cellular levels of ROS and RNS can damage DNA and promote carcinogenesis; consequently, antioxidant cellular reductants Trx and GSH, which reversibly regulate thiol modifications, have long been considered cancer-suppressing molecules [125]. They also participate in DNA synthesis and repair as electron donors for ribonucleotide reductase. Electrons required for the reduction of ribonucleotide reductase are supplied by NADPH via Trx or Grx systems [126,127]. Furthermore, they provide the counterbalancing responses that regulate proliferation and survival [3].

Finally, Trx also contributes to peroxiredoxins (Prxs) by modulating the redox status and functions as an important mediator of redox signaling [128,129]. Reduced Trx can transfer reducing equivalents to the oxidized form of Prxs, and reduced Prx in turn scavenges ROS, e.g., H_2_O_2_ [130]. Mitochondrial Prx-3 is a substrate for both Trx-2 and Grx-2 with similar catalytic efficiency via the dithiol reaction mechanism, while mitochondrial Prx-5 is limited to the Trx system [131]. These three antioxidant (Trx/Grx/Prx) systems together contribute counterbalancing responses that regulate the cellular processes of proliferation and apoptosis.

## 5. Conclusions and Future Perspectives

The thioredoxin system is known to protect cells from oxidative stress by maintaining the balance of the thiol–disulfide redox status, and its inhibition is considered a good anticancer strategy. In addition to its function as a master regulator in the redox processes, the Trx/TrxR system has received great attention over recent decades and has been implicated in vital processes such as DNA repair and synthesis, proliferation, differentiation, and apoptosis.

Indeed, in some types of cancer, the increased levels of Trx and TrxR are directly linked to aggressive tumor behavior. Current research is focused on Trx–TrxR natural and synthetic inhibitors, through covalently binding to Trx catalytic sites Cys^32^ and Cys^35^ and through binding to the Sec residue present in the active site of TrxR. In order to be useful, more in vitro and in vivo studies are nonetheless needed to elucidate the potential of Trx–TrxR inhibitors for the development of new chemotherapeutic drugs. In addition, more detailed analyses and research effort are required to understand the mechanistic roles of mitochondrial Trx-2 and TrxR-2 isoforms in tumor development and drug resistance.

## Figures and Tables

**Figure 1 biomedicines-10-01757-f001:**
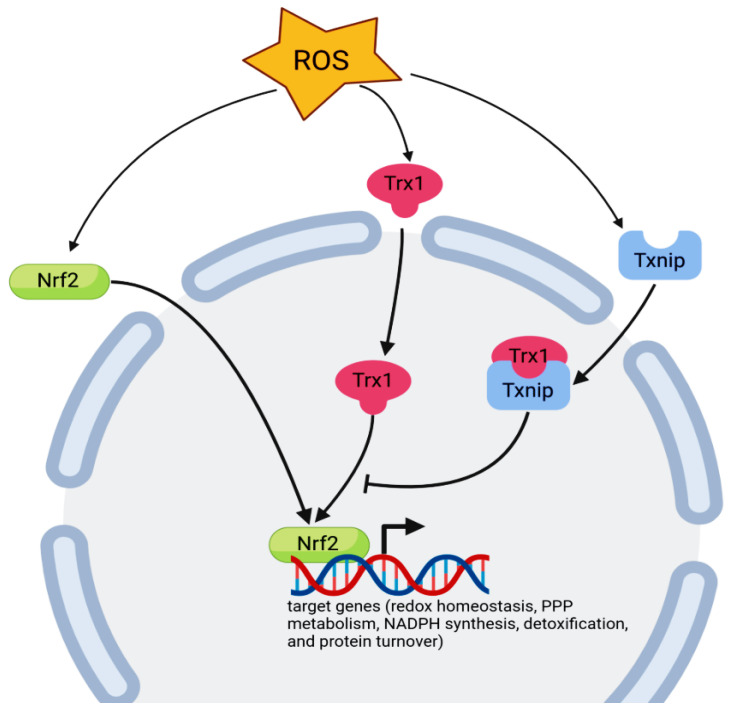
Translocation of Trx-1 and Nrf2 from the cytoplasm to the nucleus after exposure to oxidative stress. Trx1 nuclear translocation activates the transcription factor Nrf2. The activity of Trx-1 is inhibited by Txnip. Nrf2 can regulate the transcription of different target gene groups, including redox homeostasis (*NQO1*, *HO1*, *GCLC*, *GCLM*, *GSR1*, *GPX2*, *PRDX1*, *PRDX6*, *SLC7A11*, *TXN*, *TXNRD1*, *TXNIP*, and *SRX1*), pentose phosphate pathway (PPP) metabolism, NADPH synthesis (*G6PDH*, *ME1*, *PGD*, and *IDH1*), detoxification (*AKR1B3*, *GSTA1*, *GSTA2*, *GSTA3*, *GSTM1*, *GSTM2*, *GSTM3*, *GSTM4*, *GSTP1*, *PGD*, *PTGR1*, *MRP4*, and *MRP5*), and protein turnover (*PSMA1*, *PSMB5*, and *SQSTM1*) genes.

**Table 1 biomedicines-10-01757-t001:** Gold-N-heterocyclic carbine((NHC)–Au–Cl complexes and auranifin as rat TrxR inhibitors [77].

Compound	Structure	IC_50_, μM
2,3,4,6-tetra-o-acetyll-thio-b-D-glucopyrano-sato-S-(triethyl-phosphine) gold (auranofin)	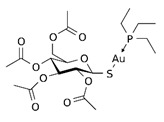	0.016
Chlorido(4-bromo-1,3-diethyl-imidazol-2-ylidene)gold(I)	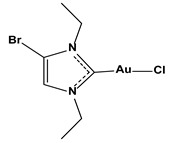	0.038
Chlorido(5-bromo-1,3-diethyl benzimidazol-2-ylidene)gold(I)	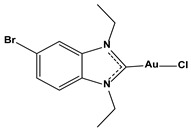	0.042
Chlorido (1,3-diethyl-5-fluoro-benzimidazol-2-ylidene)gold(I)	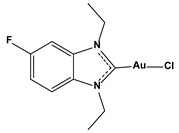	0.043
Chlorido [4-(4-bromophenyl)-1,3-diethyl-imidazol-2-ylidene]gold(I)	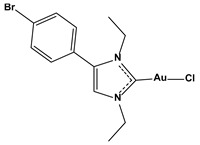	0.098
Chlorido [1,3-diethyl-4-(4-fluorophenyl)imidazol-2-ylidene]gold(I)	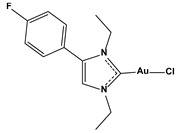	0.174
Chlorido(1,3-diethyl-imidazol-2-ylidene)gold(I)	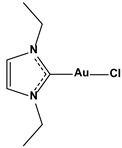	0.209
Chlorido(1,3-diethyl-4-phenylimidazol-2-ylidene)gold(I)	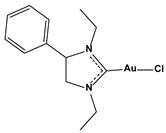	0.244

**Table 2 biomedicines-10-01757-t002:** Cytotoxicity of TrxR inhibitors.

	Chemical Class	Compound	Structure	IC_50_, μM	Cell line	Reference
1. Metal-containing inhibitors	1.1.Gold-*N*-heterocyclic carbene complexes	Bis(1,3-di(ferrocenylmethyl)imidazol-2-ylidene)-gold(I) chloride	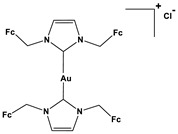 Fc = *Ferrocene*	0.14 0.19 0.12 0.48	A549 A2780 2780CP PC-3	[78]
	1.2. Gold-containing thiosugar and triethylphosphine	2,3,4,6-tetra-o-acetyll- thio-b-D-glucopyrano-sato-S-(triethyl-phosphine) gold (auranofin)	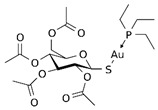	1.67 0.12 3.17	A549 HeLa MRC-5	[78] [79]
	1.3. Platinum compounds	trans-Dichlorodiamine platinum(II) (cisplatin)	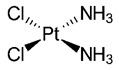	11.5 7.9	HeLa MRC-5	[80]
	1.3. Organotin-containing inhibitors	Tri-n-butyltin(IV) carboxylate	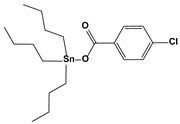	0.97 0.13	HT-29 MCF-7	[81]
	1.4. Rhodium(I) N-heterocyclic carbene complexes inhibitors	Chlorido(η^2^, η^2^-cycloocta-1,5-diene)(1,3-dimethylbenzimidazol-2-ylidene)-rhodium(I)	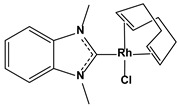	0.6 0.9	MCF-7 HT-29	[82]
2. Natural products and their synthetic analogues	2.1. Quinone compounds (Indolequinones)	5-methoxy-1-methyl-3-[(2,4,6-trifluorophenoxy)methyl]indole-4,7-dione	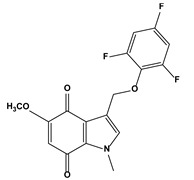	0.035 0.018	MIA PaCa-2	[83]
	2.2. Terpenoids (Atractyligenin derivative)	15-ketoatractyligenin methyl ester (SC2017)	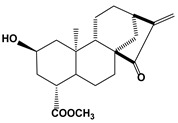	1.68	Jurkat	[84]
	2.3. Nitrosoureas and chromenes (3-nitro-2H-chromene derivative)	6-fluoro-3-nitro-2H-chromene	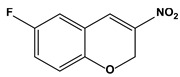	1.42	A549	[80]
	2.4. Gambogic acid	Gambogic acid	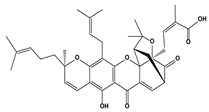	0.7	SMMC-7721	[85]
	2.5. Diketone compounds	Curcumin	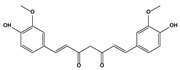	11.2 6.03 11.6 5.5 6.4	A549 H1299 H292 Tu212 Tu686	[86]
	2.6. Polyphenolic flavonoid	Quercetin	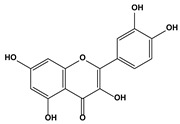	125 140	HeLa SiHa	[87]
	2.7. Colorants	Chlorophyllin	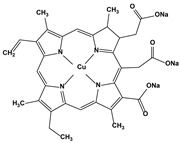	10.3 0.4 10.2 6.9 20.2	A549 HeLa HepG2 MCF7 HCT116	[88]
3. Organoselenium compounds	3.1. Organo- selenium compounds 3.2. Phenylarsenic oxide derivatives (dithiarsanes)	1,2-(5,5′-Dimethoxy(bis-1,2-benzisoselenazol-3(2*H*)-one))ethane 2-(4-Aminophenyl)-1,3,2-dithiarsenane (PAO−PDT)	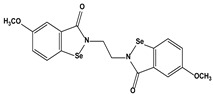 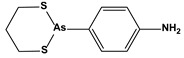	1.64 0.6	U-87MG ATCC HL-60	[89] [90]
	3.3. Ibuprofen Derivative	Phospho-ibuprofen (MDC-917)	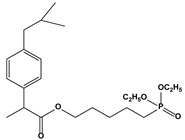	79	MCF-7	[91]

## Data Availability

Not applicable.

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
