# Peer review of "The Thioredoxin System of Mammalian Cells and Its Modulators"

_biomedicines, 2022, doi:10.3390/biomedicines10071757_

Round 1

Reviewer 1 Report

This is a high quality, comprehensive, thorough, novel, timely and up to date review article on the Trx/TrxR antioxidant system, the relevant structures and catalytic mechanisms, mainly focused on cancer biology, including current progress of Trx/TrxR inhibitors and activators in the battle against cancer. The figure and table nicely reflect the text information. The authors efficiently discussed the literature and future perspectives.

Please see below some suggestions/corrections for improving the text.

Suggestions/corrections for improving the text:

Line 16: …through…

Line 20: The phrase is a bit confusing: “The Trx/TrxR system has been used to treat cancer….” Did you mean that the Trx/TrxR system has been used as a target to treat cancer, because its function is beneficial for cancer cells? Or you wanted to say that the Trx/TrxR system is being used by cancer cells to maintain key survival proteins active through disulfide exchange reactions? Please clarify and modify the phrase accordingly.

Line 26: Connect the 2 phrases: …and highlight the factors…

Line 34: Suggested modification: …an important role is played…

Line 64: Suggested modification: …is an important protector…

Line 67: You may remove “The”: Proper oxidative protein folding...

Lines 72-73: Suggested modification: … while the structures of both oxidized and reduced Trx states from Ehrlichia chaffeensis were elucidated…

Lines 116-118: Just make the numbers superscripts: Cys62, Cys69.

Line 117: …under oxidative stress conditions?

Line 273: …in comparison with…

Table 1: Just change the comma of the first IC50 value to 0.016.

Table 2. Fix the font size in 2.5, 2.6, 2.7 (they appear a bit bigger) to match the rest of the text.

Line 279: Just add a space: …. [76]. Platinum…

Line 289: Suggested modification: ….These results indicated that the Trx/TrxR system could be…

Line 297: Just fix the superscripts: …Cys496…Sec497

Line 349: Including the Trx system…

Line 361: Superscripts just for consistent format in the text: …Ser550…Ser558

Line 378: The Trx/TrxR system…plays an important role…in regulating…

Line 418: …as electron donors…

Line 436: …the Trx/TrxR system…

Line 439: …levels of…

Author Response

Dear reviewer. We are highly grateful for your consideration of this manuscript and its appreciation. Thank you very much for the valuable comments and suggestions that we used for the correction of the manuscript.

The manuscript has been corrected in English by MDPI English Editing Service.

In according to your suggestions, the following corrections has been made to the manuscript:

line 16: … have significant role in cell homeostasis through redox signaling…

line 20. We agree with your remark and the phrase “Trx/TrxR system has been used to treat cancer” has been corrected to “The Trx/TrxR system has been used as a target to treat cancer….”

line 26. The 2 phrases have been connected - “Herein we review the structure and catalytic properties of the Trx/TrxR system, its role in cellular signaling in connection with other redox systems and the factors that modulate the Trx system”.

line 35.  The correction has been made - from “…the important role is played by the thioredoxin (Trx)-dependent…” to “…an important role is played by the thioredoxin (Trx)-dependent…”.

line 65. The correction has been made - from “…In mammals, the Trx system is the important protector…” to “…In mammals, the Trx system is important for protection …”.

line 68 The correction has been made - from “The proper oxidative protein folding within cells..” to “Proper oxidative protein folding within cells..”.

lines 74-75. The correction has been made - from “…the structures of both oxidized and reduced states of Ehrlichia chaffeensis were elucidated by NMR in solution” to “…the structures of both oxidized and reduced Trx states from Ehrlichia chaffeensis have been elucidated by NMR in solution…”

lines 117-118. The correction has been made - from “…for example, the non-active site disulfide formed between Cys62 and Cys69 inhibits Trx-1 activity for redox signaling under oxidative.” to “…for example, the non-active site disulfide formed between Cys62 and Cys69 inhibits Trx-1 activity for redox signaling under oxidative stress conditions.”

line 273. The correction has been made - from” … but less effective in compare with auranofin.” to “…but less effective in comparison with auranofin.”

Table 1. The comma of the first IC50 value has been changed to 0.016.

Table 2. The font size in 2.5, 2.6, 2.7 has been fixed.

line 279: The correction has been made - from “…in cancer cells [76].Platinum-containing drugs…” to “…in cancer cells [76]. Platinum-containing drugs” (A space has been added).

line 289. The correction has been made - from “At the same time Trx system could be a target…” to “These results indicated that the Trx system could be a target for the toxicity of MeHg…”

line 297. The correction has been made - from “…active residues Cys496 and Sec497 in TrxR” to “…active residues Cys496 and Sec497 in TrxR”.

line 349. The correction has been made - from “…including Trx system,…” to “…including the Trx system,…”

line 361. The correction has been made - from “…phosphorylates Nrf2 at Ser550 in vivo and at Ser558 residue…” to “…phosphorylates Nrf2 at Ser550 in vivo and at Ser558 residue…”

line 377. The correction has been made - from “Trx/TrxR system plays the important role in regulation of Nrf2 activity” to “The Trx/TrxR system plays an important role in regulation of Nrf2 activity”

line 417. The correction has been made - from “…repair as an electron donor for ribonucleotide reductase.” to “…repair as electron donors for ribonucleotide reductase.”

line 434. The correction has been made - from “…Trx/TrxR system has received great attention …” to “…the Trx/TrxR system has received great attention…”

line 437. The correction has been made - from “…the increased level sof Trx and TrxR…” to “…the increased levels of Trx and TrxR…”

Thank you again for your valuable comments.

Dear reviewer. We are highly grateful for your consideration of this manuscript and its appreciation. Thank you very much for the valuable comments and suggestions that we used for the correction of the manuscript.

The manuscript has been corrected in English by MDPI English Editing Service.

In according to your suggestions, the following corrections has been made to the manuscript:

line 16: … have significant role in cell homeostasis through redox signaling…

line 20. We agree with your remark and the phrase “Trx/TrxR system has been used to treat cancer” has been corrected to “The Trx/TrxR system has been used as a target to treat cancer….”

line 26. The 2 phrases have been connected - “Herein we review the structure and catalytic properties of the Trx/TrxR system, its role in cellular signaling in connection with other redox systems and the factors that modulate the Trx system”.

line 35.  The correction has been made - from “…the important role is played by the thioredoxin (Trx)-dependent…” to “…an important role is played by the thioredoxin (Trx)-dependent…”.

line 65. The correction has been made - from “…In mammals, the Trx system is the important protector…” to “…In mammals, the Trx system is important for protection …”.

line 68 The correction has been made - from “The proper oxidative protein folding within cells..” to “Proper oxidative protein folding within cells..”.

lines 74-75. The correction has been made - from “…the structures of both oxidized and reduced states of Ehrlichia chaffeensis were elucidated by NMR in solution” to “…the structures of both oxidized and reduced Trx states from Ehrlichia chaffeensis have been elucidated by NMR in solution…”

lines 117-118. The correction has been made - from “…for example, the non-active site disulfide formed between Cys62 and Cys69 inhibits Trx-1 activity for redox signaling under oxidative.” to “…for example, the non-active site disulfide formed between Cys62 and Cys69 inhibits Trx-1 activity for redox signaling under oxidative stress conditions.”

line 273. The correction has been made - from” … but less effective in compare with auranofin.” to “…but less effective in comparison with auranofin.”

Table 1. The comma of the first IC50 value has been changed to 0.016.

Table 2. The font size in 2.5, 2.6, 2.7 has been fixed.

line 279: The correction has been made - from “…in cancer cells [76].Platinum-containing drugs…” to “…in cancer cells [76]. Platinum-containing drugs” (A space has been added).

line 289. The correction has been made - from “At the same time Trx system could be a target…” to “These results indicated that the Trx system could be a target for the toxicity of MeHg…”

line 297. The correction has been made - from “…active residues Cys496 and Sec497 in TrxR” to “…active residues Cys496 and Sec497 in TrxR”.

line 349. The correction has been made - from “…including Trx system,…” to “…including the Trx system,…”

line 361. The correction has been made - from “…phosphorylates Nrf2 at Ser550 in vivo and at Ser558 residue…” to “…phosphorylates Nrf2 at Ser550 in vivo and at Ser558 residue…”

line 377. The correction has been made - from “Trx/TrxR system plays the important role in regulation of Nrf2 activity” to “The Trx/TrxR system plays an important role in regulation of Nrf2 activity”

line 417. The correction has been made - from “…repair as an electron donor for ribonucleotide reductase.” to “…repair as electron donors for ribonucleotide reductase.”

line 434. The correction has been made - from “…Trx/TrxR system has received great attention …” to “…the Trx/TrxR system has received great attention…”

line 437. The correction has been made - from “…the increased level sof Trx and TrxR…” to “…the increased levels of Trx and TrxR…”

Thank you again for your valuable comments.

Reviewer 2 Report

Plesae see attached document.

Author Response

Dear reviewer. We are highly grateful for your consideration of this manuscript. Thank you very much for the valuable comments and suggestions that we used for the correction of the manuscript.

According to your suggestion, we would like to replace the title of the review to “The thioredoxin system of mammalian cells and its modulators” focusing on the most interesting facts about inhibitors and activators.

In according to your suggestions, the following corrections also has been made to the manuscript:

line 20. We agree with your remark and the phrase “Trx/TrxR system has been used to treat cancer” has been corrected to “The Trx/TrxR system has been used as a target to treat cancer….”

line 22. The correction has been made from “The TrxR keeps reduced Trx level using NADPH as a cofactor…” to “The TrxR keeps reduced Trx level using NADPH as a co-substrate…”

line 26. The 2 phrases have been connected: “Herein we review the structure and catalytic properties of the Trx/TrxR system, its role in cellular signaling in connection with other redox systems, and the factors that modulate the Trx system”.

line 43. The correction has been made from “The Trx-dependent system includes disulfide reductase Trx and thioredoxin reductase (TrxR)” to “The Trx-dependent system includes disulfide reductase Trx and thioredoxin reductase (TrxR) used NADPH(H+) as a co-substrate

line 44-45. The correction has been made from “…peroxiredoxins, which catalyze the decomposition of H2O2…” to “…peroxiredoxins, which catalyze the reduction of H2O2 to water …”

line 45. The correction has been made from “…the mechanism for controlling the cellular reactive oxygen species (ROS) level.” to “…the mechanism of controlling the cellular reactive oxygen species (ROS) level.”

line 45. The numbers have been made superscripts (…Cys32 and Cys35….) here and on lines 114-115, 298, 362-363.

line 53. The correction has been made from “The overexpression of both cytoplasmic and mitochondrial TrxR isoforms (TrxR1, TrxR2) was found in various types of malignant neoplasms” to “The overexpression of both cytoplasmic and mitochondrial TrxR isoforms (TrxR1, TrxR2) has been found in various types of malignant neoplasms”.

line 64. The correction has been made from “In mammals, the Trx system is an important protector from the effects of ROS [12].” to “In mammals, the Trx system is important for protection from the effects of ROS [12]”

line 78. The correction has been made from “…surrounding the central core of a four-stranded β-sheet” to “…surrounding a central core made of four β-sheets”.

line 88. Reference has been inserted to the sentence: In mammalian cells, two major Trx types have been characterized that differ in their intracellular localization, tissue-specific expression patterns and subcellular structure:  cytosolic Trx-1 and mitochondrial Trx-2 [18].

lines 109-111. The correction has been made - from “The oxidized Trx must then be reduced back to its active state by accepting electrons from NADPH(H+) in a reaction catalyzed by TrxR, enabling the onset of the next reaction cycle” to “Oxidized Trx is reduced to its active state by electrons from NADPH(H+). The reaction is catalyzed by TrxR and enables the onset of the next reaction cycle”.

lines 116-120. The correction has been made – from “These additional cysteine residues are involved in the activity of mammalian Trx depending on their redox state; for example, the non-active site disulfide formed between Cys62 and Cys69 inhibits Trx-1 activity for redox signaling under oxidative S-nitrosylation of Trx at Cys69…” to “These additional cysteine residues are involved in the activity of mammalian Trx depending on their redox state; for example, the non-active site disulfide formed between Cys62 and Cys69 inhibits Trx-1 activity for redox signaling under oxidative stress conditions. S-nitrosylation of Trx at Cys69 ….”.

line 126. The correction has been made – from “…the galactose receptor folding in E.coli” to “…the galactose receptor folding in E. coli”.

line 142.  The correction has been made – from “…identified primarily in mammalians…” to “…identified primarily in mammals…”.

line 163. The correction has been made – from “…which has properties characteristic of the mitochondrial translocation signal” to “…which has the characteristic properties of the mitochondrial translocation signal”.

line 180. The correction has been made – from “…the extracellular environment via ER/Golgi-independent pathway…” to “…the extracellular environment via the ER/Golgi-independent pathway…”.

line 199. The correction has been made – from “…which Cys35 of the active site is replaced with serine, was quickly bound to the cell surface…” to “…which Cys35 of the active site was replaced with serine, was quickly bound to the cell surface…”.

Pages 7 -10. Table 1: gold-Au-Cl complexes are presented as inhibitors of TrxR purified from rat liver (Sigma-Aldrich).

Table 2: It has been reduced the figure size of compound 2.4 and corrected the name of compound 3.1. The typo has been deleted (0.7).  

lines 301-304. The correction has been made – from “Myricetin and quercetin have a strong anticancer activity by irreversible inhibition of TrxR via an attack on the reduced C-terminal -Cys-Sec-Gly- active site of TrxR, in a time-, NADPH-, oxygen-, and concentration-dependent manners, ultimately leading to cell death” to “Myricetin and quercetin irreversibly inhibit TrxR (IC50 values of 0.62 and 0.97 μmol/L, respectively) and arrest the growth of a lung cancer cell line. The inhibition of TrxR was related to time exposure to the inhibitors, the concentration of NADPH(H+), and the amount of dissolved oxygen [88].”

lines 354-358. The correction has been made – from “Phosphorylation of Nrf2 is an integral component of its activation necessary for the nuclear localization and transcription activation of different target gene groups including genes encoding antioxidant proteins and enzymes. Among them glutathione peroxidase 1 (GPX1), glutathione S-transferase mu 1 (GSTM1), glutamate-cysteine ligase catalytic subunit (GCLC); glutathione reductase (GSR); ferrochelatase (FECH); TRX, TXNRD1, NAD(P)H quinone dehydrogenase 1 (NQO1)] [100].” to “Active phosphorylated Nrf2 is transferred to the nucleus to affect the transcription of genes encoding antioxidant proteins and enzymes such as glutathione peroxidase 1 (GPX1), glutathione S-transferase mu 1 (GSTM1), glutamate–cysteine ligase catalytic subunit (GCLC); glutathione reductase (GSR); ferrochelatase (FECH); TRX, TXNRD1, and NAD(P)H quinone dehydrogenase 1 (NQO1)] [100].”

line 387. The correction has been made – from “…controlling cell fate by Trx system…” to “…controlling cell fate by the Trx system…”.

lines 432-434. The correction has been made – from “For a long time, the Trx/TrxR system plays the most prominent role in protecting the cell from oxidative stress by maintaining the balance of the thiol-disulfide redox status, remaining at the forefront of research of mechanisms of cancer spread.” to “The thioredoxin system is known to protect cells from oxidative stress by maintaining the balance of the thiol–disulfide redox status, and its inhibition is considered a good anticancer strategy.

 line 438. The correction has been made – from “…in some cancers…” to “…in some types of cancer…”.

line 438. The correction has been made - from “…level sof Trx and TrxR…” to “…levels of Trx and TrxR…”

Thank you again for your valuable comments and suggestions.